# Investigation and Identification of Cyst Nematodes in theBashang Region of Hebei, China

Yuhuan Wu [1,2,3], Huan Peng [3], Shiming Liu [3], Hudie Shao [3], Yunqing Li [3], Yingdong Zhang [3], Yaning Li [1], Daqun Liu [1,*] and Deliang Peng [3]

[1] College of Plant Protection, Technological Innovation Center for Biological Control of Crop Disease and Insect Pests of Hebei Province, Hebei Agricultrual University, Baoding 071000, China

[2] College of Agricultural and Forestry Science and Technology, Hebei North University, Zhangjiakou 075000, China

[3] State Key Laboratory for Biology of Plant Diseases and Insect Pests, Institute of Plant Protection, Chinese Academy of Agricultural Sciences, Beijing 100193, China

* Correspondence: liudaqun@caas.cn

**Abstract:** Cyst nematodes are one of the most important pathogens worldwide. Most cyst nematode species have been reported recently in China. From 2016 to 2020, an extensive survey of cyst nematodes was conducted in the Bashang region of Hebei Province. A total of 158 soil samples were collected, and cyst-forming nematodes were isolated from five soil samples. Morphological and molecular characterization showed that four of the cyst-forming nematode populations were *Heterodera glycines* (SCN), named populations of ZM, KM, CB and FN, respectively. These SCN populations were collected from Zhangbei County, Kangbao County, Chabei Management Area of Zhangjiakou and Fengning Manchu Autonomous County of Chengde, respectively, where the corresponding cyst densities were 57, 41, 103 and 31 cysts/200 cc soil. Furthermore, the populations of ZM, KM and CB were identified as race 4, whereas the FN population was identified as race 3. The cyst-forming nematode population was collected from Zhangbei County of Zhangjiakou, which was confirmed to be *Heterodera schachtii* (SBCN), named population ZZ, and the cyst density was 94 cysts/200 cc soil. It is a new disease of Chinese cabbage caused by SBCN based on Koch's postulates. Fourteen cultivars from five plant families were evaluated as hosts for SBCN. Chinese cabbage (cv. Linglonghuang012) and cabbage (cv. Chunwang) were suitable hosts, while celery (cv. Yuhuang), potato (cv. Helan 15) and eggplant (cv. Junlang) were nonhosts. The obtained results regarding the occurrence, distribution, races of SCN and hosts of SBCN in the Bashang region in this study provide a reference for SCN and SBCN management.

**Keywords:** *Heterodera glycines*; *Heterodera schachtii*; race; host; Bashang region

## 1. Introduction

Cyst nematodes are one of the most destructive pests worldwide. Soybean cyst nematode (*Heterodera glycines*, SCN) is the most important pest on soybean, and it causes huge losses in yield annually. Soybean cyst nematodes are widely distributed in many provinces in China, such as Heilongjiang, Jilin, Liaoning, Inner Mongolia, Beijing, Henan, Hebei, Shandong, Shanxi, Anhui, Jiangsu, Hubei, Zhejiang, Shanghai, Xinjiang, Shanxi, Ningxia, Gansu, Guangxi, Guizhou and Jiangxi [1–5]. In Hebei Province, soybean cyst nematodes were detected in Handan, Xingtai, Baoding, Shijiazhuang, Qinhuangdao, Tangshan, Cangzhou and Zhangjiakou [6,7]. In 1962, the physiological differentiation of SCN was first reported [8]. In 1970, a procedure for differentiating SCN races using four differentials and a susceptible standard was established [9]. Eleven additional races were first described, and later, this expanded the total to 16 SCN races [10]. There are SCN races 1, 2, 3, 4, 5, 6, 7, 9 and 14 in China, but SCN races 1, 3 and 4 are most widely distributed [7,11,12]. Among them, SCN races 1, 2, 4, 5, 7 and 14 were reported in Shandong Province [11,13].

SCN races 1, 2, 4, 5 and 9 were found in Henan Province [7]. SCN races 1, 3 and 6 were reported in Liaoning Province [14], and SCN race 4 was reported in Beijing [15]. In Hebei Province, SCN races 1, 2 and 5 have been found in Handan, Gaocheng and Cangxian, respectively [7].

Sugar beet cyst nematode (*Heterodera schachtii*, SBCN) is the most economically important pest in sugar beet production areas and is listed as a quarantine nematode in China. It was first found on sugar beet in Germany in 1859. Recently, it was reported in many countries and regions [16–20]. SBCN causes considerable damage to many crops including Chinese cabbage in South Korea [21], cauliflower in Jordan [22], Brassica vegetables in Japan [23], broccoli in the USA [24] and sugar beet in Xinjiang, China [20]. SBCN reproduces on a wide range of plants, including Chenopodiaceae, Cruciferae and Leguminosae [3,20,25].

However, in the Bashang region of Hebei Province, information on the occurrence and distribution of cyst nematodes is limited. Therefore, a wide investigation of cyst nematodes in the Bashang region of Hebei Province is urgent. In this study, the cyst nematode distribution in the Bashang region of Hebei Province was surveyed, and its characterization was analyzed based on morphological and molecular methods. Subsequently, SCN races were identified through phenotyping by the female index, and host ranges of SBCN were conducted in pot experiments. The results obtained in this study will provide a scientific basis for taking effective measures to control SCN and SBCN, and the identified predominant races will be beneficial for breeding programs, in the Bashang region.

## 2. Materials and Methods

### 2.1. Collection and Isolation of Cyst Nematodes

A total of 158 soil samples were collected from the Bashang region of Hebei Province during 2016–2020 (Table S1). In each plot, the soil at five sampling sites was arbitrarily collected from the root zone at a depth of 5–20 cm and thoroughly mixed after removing the top 0–5 cm layer of surface soil. Each soil sample was placed in a plastic bag, transported to the laboratory, and stored at 4 °C for further analyses.

For nematode extraction, large soil clumps were crushed by hand, and each soil sample was well mixed. Two hundred milliliters of soil was used for cyst isolation using Cobb's sieving decanting method [3]. The cysts were handpicked under a stereomicroscope (Olympus, SZ61, Tokyo, Japan). A tray method was used to extract the second-stage juveniles (J2s) and males [26]. Eggs were extracted from the collected cysts, which were broken on a 72-μm-apeture sieve and collected on a 25-μm-apeture sieve. The cysts were placed in a sterile vial containing 0.5% sodium hypochlorite for 3 min, washed with sterile water three times, placed in a 3 mmol/L $ZnCl_2$ solution, and incubated in the dark at 25 °C. Then, J2s were collected.

### 2.2. Morphological Characterization

All the nematodes were killed in water at 65 °C for 3 min and fixed in thermal 4% formalin for 30 min. Samples were preserved in permanent slides for morphological analyses [27]. SCN and SBCN populations were identified based on permanent slides of the vulval cone regions of cysts and of the measurements of main characters of J2s. The vulva cones prepared after dissection were cleaned and trimmed. Morphometrics were measured under a light microscope (Olympus, SZX7, Tokyo, Japan).

### 2.3. Molecular Identification

DNA was extracted from a single cyst using the method described in Ou et al. [28]. Then, 3 μL proteinase K solution (600 ng/mL) and 7 μL 10× PCR buffer (Takara Bio, Shiga, Japan) were added to the PCR tube, and the mixture was placed in liquid nitrogen for 2 min. The cyst was crushed thoroughly with a glass rod, and then the PCR tube was incubated at 65 °C for 1 h and 85 °C for 15 min. The supernatant DNA suspension for PCR amplification was stored at −20 °C.

The primer pairs TW81 and AB28 (Table 1) were used for amplification of the internal transcribed spacer. The PCR amplification conditions were as follows: an initial denaturation at 94 °C for 5 min, followed by 35 cycles of denaturation at 94 °C for 30 s, annealing at 55 °C for 45 s and extension at 72 °C for 1.5 min with a final extension step at 72 °C for 10 min. The obtained new sequences were deposited in GenBank, and the phylogenetic tree was constructed using MEGA 6.0 (Center for Evolutionary Medicine and Informatics, Biodesign Institute, Tempe, AZ, USA) with the maximum likelihood model [29].

**Table 1.** Codes and sequences of primers used in this study.

| Primer Name | Sequence 5′-3′ | Types | Reference |
|---|---|---|---|
| TW81 | 5′-GTTTCCGTAGGTGAACCTGC-3′ | rDNA-ITS universal primers | [30] |
| AB28 | 5′-ATATGCTTAAGTTCAGCGGGT-3′ | | |
| SCNFI | 5′-GGACCCTGACCAAAAAGTTTCCGC-3′ | SCAR primers | [28] |
| SCNRI | 5′-GGACCCTGACGAGTTATGGGCCCG-3′ | | |
| OPA06-HsF | 5′-GGACCCTGACGACCAGAATA-3′ | SCAR primers | [31] |
| OPA06-HsR | 5′-GACAACACGAAGGAGCGAGC-3′ | | |
| D2A | 5′-ACAAGTACCGTGAGGGAAAGTTG-3′ | 28S-rDNA universal primers | [32] |
| D3B | 5′-TCGGAAGGAACCAGCTACTA-3′ | | |

Four populations (ZM, KM, CB and FN) of *H. glycines* and the ZZ population of *H. schachtii* isolated in this study were used; moreover two SCN populations (Hg1 and Hg2) collected from Langfang City of Hebei Province and Henan Province [28] and three SBCN populations (Hs1, Hs2 and Hs3) from Xinjiang [31] were used as positive controls. These ten populations of cyst nematodes were used for primer specificity analysis. The PCR amplification reaction was carried out as follows: The 25 μL PCR mixture contained 12.5 μL 2× PCR buffer for KOD FX (Toyobo Life Science Co., Ltd., Tokyo, Japan), 5 μL dNTPs (2 mM), 1 μL of each specific primer at 10 μM (SCNFI/SCNRI and OPA06-HsF/OPA06-HsR), 1 μL of each pair of universal primers at 10 μM (D2A/D3B) (Table 1), 1 μL DNA, 0.5 μL KOD FX DNA polymerase (1.0 U/μL), and 4 μL distilled water. The thermocycler was programmed following conditions: 94 °C for 4 min; 35 cycles of 94 °C for 1 min, 56 °C for 1 min, and 72 °C for 2 min; followed by 72 °C for 10 min. The PCR products were separated on a 2% agarose gel, visualized by Gelred gel staining (GelStain, TransGen Biotech, Beijing, China), and photographed under UV light.

*2.4. Identification of SCN Races*

Seeds of four differentials ('Pickett', 'Peking', 'PI88788', 'PI907653') and Lee were planted into 7.5 cm plastic pots containing a mixture of sterilized sand and soil (*v:v* = 7:3). Each seedling was inoculated with 1500 J2s of SCN at 15 d. The seedlings were grown at 28 °C. At 28 days post-inoculation (dpi), the plants were uprooted, and cysts were collected with sieves of 850-μm-apeture and 150-μm-apeture and counted under a stereomicroscope (Olympus, SZ61, Japan). The plant was classified as susceptible "+" when the female index (FI) $\geq$ 10% and as resistant "-" when FI < 10%. The race type was determined by the FI value according to Riggs and Schmmit [10]. Female index (FI, %) = (Average number of cysts from differential/Average number of cysts from Lee) $\times$ 100.

*2.5. Koch's Postulates Test and the Host Range Experiment of the SBCN*

Fourteen plant cultivars were planted into 11 cm plastic pots containing a mixture of autoclaved sand and soil (*v:v* = 7:3). Each seedling was inoculated with 800 J2s from the obtained SBCN population. The seedlings were grown at 25 °C under a photoperiod of 16 h light/8 h dark [33]. After 3, 8, 15, 30 and 45 days of incubation, the roots were washed and stained using acid fuchsin based on the dyeing method [34]. The cysts in the soil were

isolated and counted under a light microscope (Olympus, SZX7, Tokyo, Japan) and were detected by the molecular method described above.

## 3. Results

### 3.1. Isolation of Cyst Nematodes

From 2016 to 2020, a total of 158 soil samples were collected from 8 counties in the Bashang region of Hebei Province. Among the collected samples, cyst-forming nematodes were isolated from five soil samples. Of them, the cyst density of the ZM population collected from Lisenlin Village, Zhangbei County of Zhangjiakou; the KM population collected from Dasiduan Village, Kangbao County of Zhangjiakou; the CB population collected from the Chabei Management District of Zhangjiakou; the FN population collected from Fengning Manchu Autonomous County of Chengde; and the ZZ population collected from Yongshengchang Village, Zhangbei County of Zhangjiakou reached 57 cysts/200 cc soil, 41 cysts/200 cc soil, 103 cysts/200 cc soil, 31 cysts/200 cc soil and 94 cysts/200 cc soil, respectively (Table 2).

**Table 2.** Density of cysts in soil samples collected from the Bashang region of Hebei Province.

| Location | Total Number of Plot Inspected | Number of Plot Inspected | Population | Density of Cysts (NO./200 cc soil) |
|---|---|---|---|---|
| Zhangbei county | 94 | 1 | ZM | 57 |
| | | 1 | ZZ | 94 |
| Kangbao county | 17 | 1 | KM | 41 |
| Guyuan county | 11 | 0 | - | - |
| Shangyi county | 11 | 0 | - | - |
| Chabei management district | 4 | 1 | CB | 103 |
| Saibei management district | 4 | 0 | - | - |
| Weichang manchu autonomous county | 12 | 0 | - | - |
| Fengning manchu autonomous county | 5 | 1 | FN | 31 |

"-" denotes that no cysts were collected in this study.

### 3.2. Morphological Identification

The morphological and morphometrical characteristics of the five populations were evaluated under a light microscope. The morphology observation of populations (ZM, KM, CB and FN) showed that cysts were lemon-shaped with a protruding neck and vulval cone, ambifenestrate, finger-like bullae prominent, and well-developed underbridge, the J2s were vermiform with a hyaline region in the tail terminus, and eggs were oval (Figure 1A–F). The key morphometrics of cysts (*n* = 20) are listed in Table S2. The morphology of the cysts and juveniles of populations (ZM, KM, CB and FN) were typical of *H. glycines*.

Flask-shaped cysts of the ZZ population were light to dark brown and ambifenestrate with strongly developed molar-shaped bullae and underbridges (Figure 1G–L). The key morphometrics of cysts (*n* = 20) were as follows: a mean cyst length excluding neck of 831.71 ± 49.41 (700.83 to 874.96) μm; cyst width 476.42 ± 67.77 (384.40 to 493.45) μm; vulval slit 45.12 ± 3.26 (40.83 to 51.38) μm, and underbridge length 97.09 ± 8.74 (77.85 to 112.73) μm. Second-stage juveniles (*n* = 20): body length 435.73 ± 11.21 (401.88 to 444.32) μm, body width 21.28 ± 0.54 (20.37 to 22.78) μm, stylet length 24.83 ± 0.46 (22.78 to 24.59) μm, tail length 48.25 ± 2.25 (44.98 to 53.75) μm, and hyaline length 34.82 ± 2.71 (32.12 to 35.80) μm. The morphology of the cysts and juveniles were typical of *H. schachtii* (Table S3).

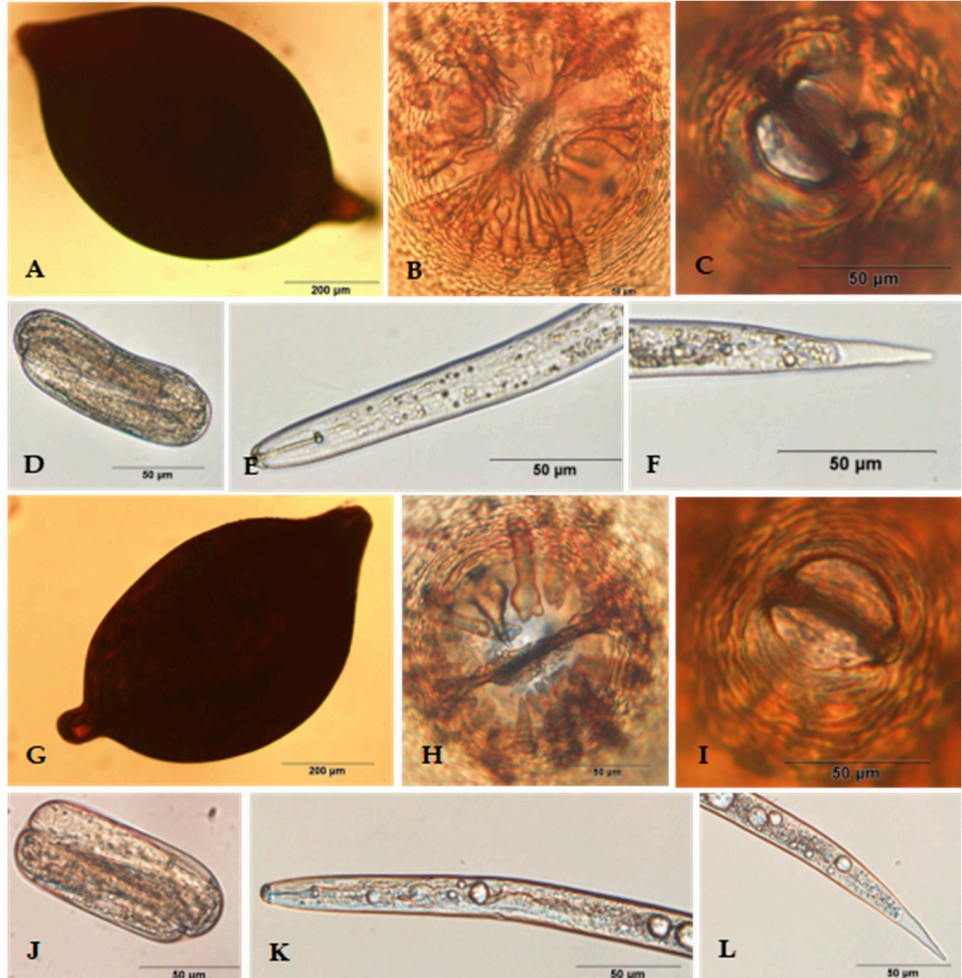

**Figure 1.** Morphological characteristics of SCN (**A**–**F**) and SBCN (**G**–**L**). (**A**) and (**G**) Cyst; (**B**) and (**H**) Bullae; (**C**) and (**I**) Ambifenestrate; (**D**) and (**J**) Egg; (**E**) and (**K**) Stylet; (**F**) and (**L**) Hyaline region.

*3.3. Molecular Identification*

The ITS regions were amplified from individual cysts and sequenced. Amplicons of 971 bp, 972 bp, 972 bp, 966 bp and 970 bp were produced for the five populations. The obtained sequences were deposited in GenBank (GenBank Accession Nos. MT199662-MT199665 and ON644871). The phylogenetic analyses using the ITS sequences showed that populations of ZM, KM, CB and FN were clustered in a well-supported clade and closely related to *H. glycines* populations (GU595432.1, KY794756.1 and KU160508.1). The BLAST results of ZZ population sequence confirmed that it was 99.79–100% similar to those of *H. schachtii* from XJ11-5 (MW856648), XJ11-H2 (MW856649) and XJ8-2 (MW856650). The results indicated that *H. schachtii* of the ZZ population was clustered with South Korea (MF043911.1), South Africa (MF754150.1), Belgium (AY166438.1), XJ11-5 (MW856648), the USA (EF611102.1), France (EF611103.1), XJ11-H2 (MW856649), XJ8-2 (MW856650) and Belgium (EF611105.1) within a group at a value of 100% (Figure 2).

The specificity of the assay was evaluated using specific species primers of SCN and SBCN. The results showed that 477 bp fragments were obtained from the ZM, KM, CB and FN populations and two positive controls, and positive bands of 922 bp fragments were observed from the ZZ and Xinyuan populations (Figure 3A,B). The DNA quality of all cysts was examined using a universal primer (D2A/D3B), and the results indicated that all DNA quality was high (Figure 3C).

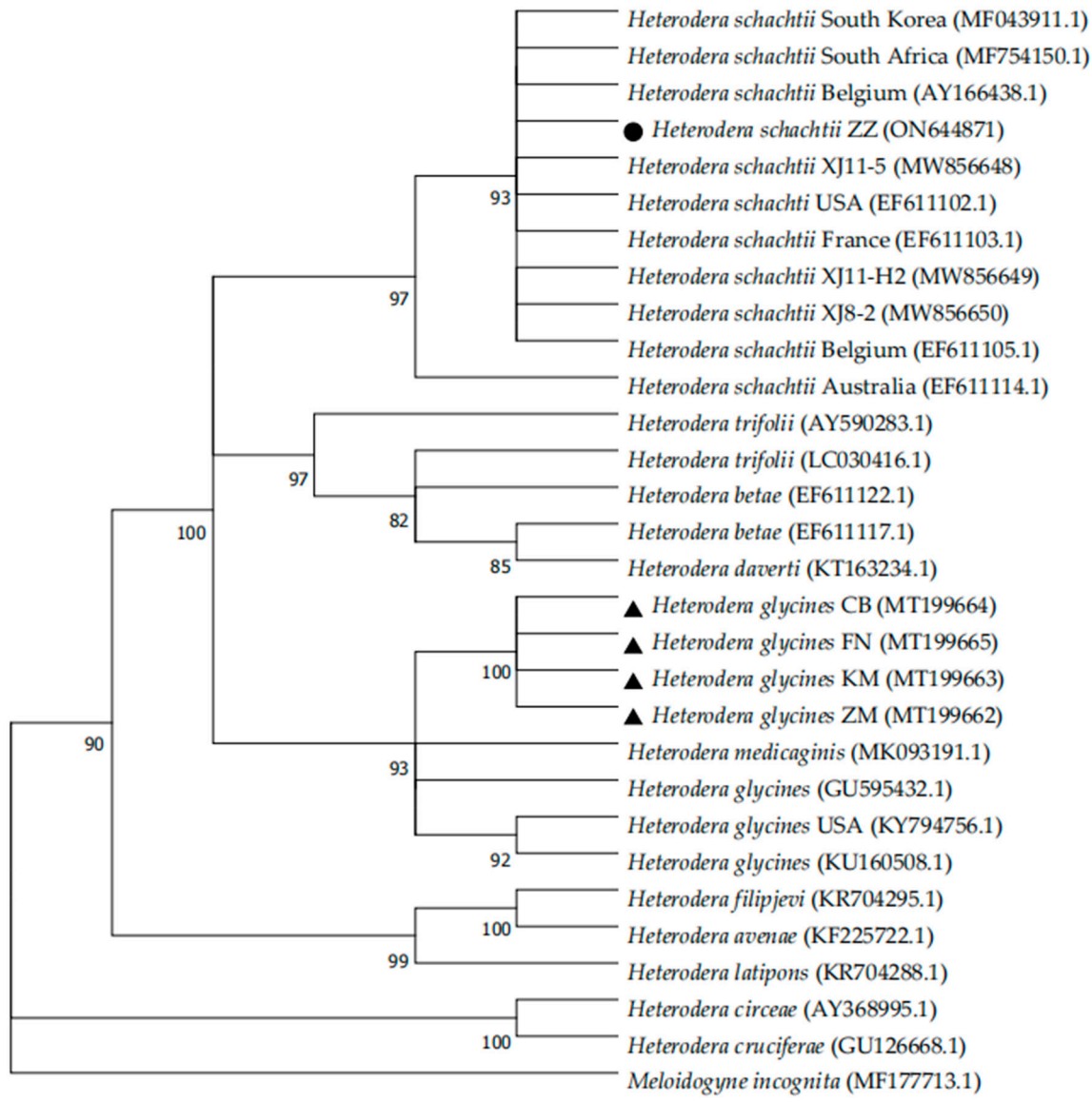

**Figure 2.** Phylogenetic tree of cyst nematodes collected from Bashang and other Heterodera species based on the rDNA-ITS sequences by using the maximum likelihood model. Meloidogyne incognita was selected as the outgroup. Numbers are bootstrap values.

### 3.4. Identification of SCN Races

Four differentials ('Pickett', 'Peking', 'PI88788', 'PI907653') and the control 'Lee' were used to identify the races of SCN according to the phenotypes using the female index. The results showed that compared to the control Lee, the female index of Peking, Pickett, PI88788 and PI90763 infected with the ZM population was 86.4%, 77.9%, 75.4% and 71.8%, respectively, which were all higher than 10%; hence, the ZM population was identified to be SCN race 4; The female index of Peking, Pickett, PI88788 and PI90763 infected with the KM population was 61.7%, 66.9%, 45.6% and 91.9%, respectively. Therefore, the KM population was identified to be SCN race 4, and the female index of the four differentials infected with the CB population was 81.5%, 60.0%, 54.5% and 68.5%, respectively, so the CB population was also SCN race 4. However, the female index of Peking, Pickett, PI88788 and PI90763 infected with the FN population was 0.3%, 4.1%, 7.9% and 1.7%, respectively, which were all less than 10%; thus, the FN population was SCN race 3 (Table 3).

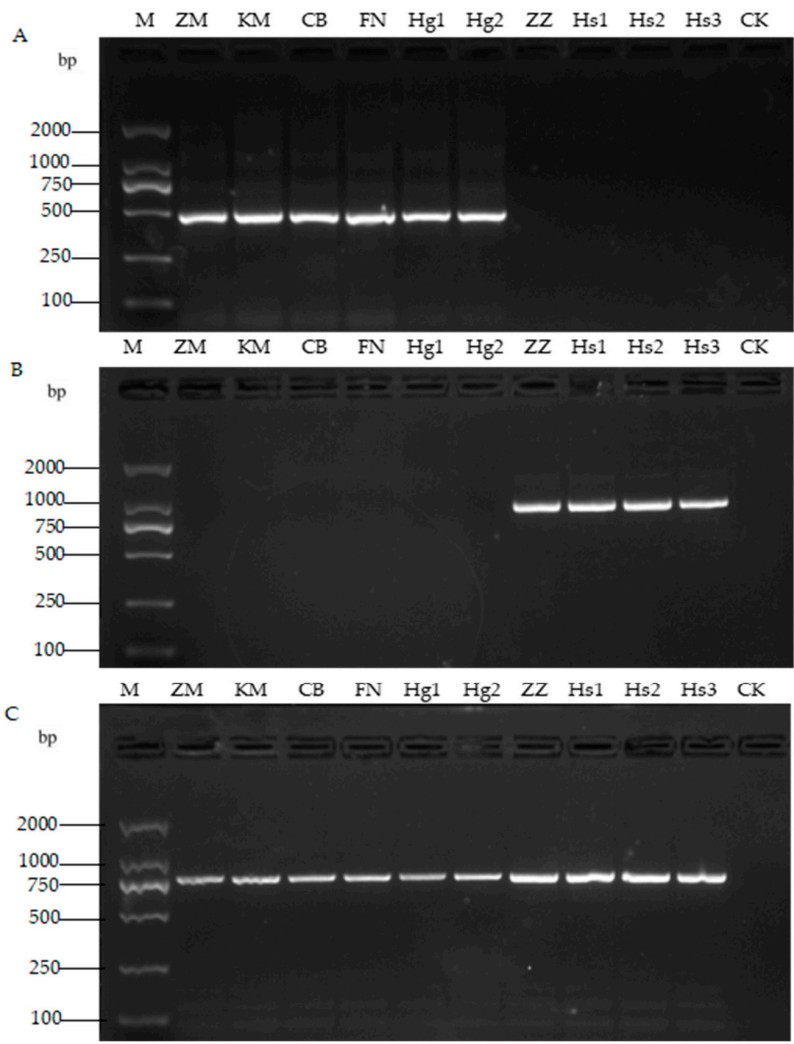

**Figure 3.** Specificity assessment of the SCAR marker for the detection of SCN and SBCN. PCR products from ten cyst nematode populations using SCAR-PCR. (**A**) A total of 477 bp bands were observed from six populations of *H. glycines* (ZM, KM, CB, FN, Hg1 and Hg2). (**B**) A 922 bp band was obtained from four populations of *H. schachtii* (ZZ and Hs1-Hs3). (**C**) PCR results of D2A/D3B primers. CK, negative control (ddH$_2$O instead of DNA). M, D2000 DNA ladder.

**Table 3.** Identification of *Heterodera glycines* races collected from the Bashang region of Hebei Province.

| Populations | Survey Item | Soybean Differential Hosts | | | | | Race |
|---|---|---|---|---|---|---|---|
| | | **Peking** | **Pickett** | **PI88788** | **PI90763** | **Lee** | |
| ZM | No. of cysts | 124.4 ± 9.8 | 112.2 ± 11.3 | 108.6 ± 11.1 | 103.4 ± 4.8 | 144.0 ± 10.2 | 4 |
| | FI (%) | 86.4 | 77.9 | 75.4 | 71.8 | 100 | |
| | Reaction | + | + | + | + | | |
| KM | No. of cysts | 77.6 ± 5.1 | 84.2 ± 4.3 | 57.4 ± 5.5 | 115.6 ± 7.2 | 125.8 ± 5.9 | 4 |
| | FI (%) | 61.7 | 66.9 | 45.6 | 91.9 | 100 | |
| | Reaction | + | + | + | + | | |
| CB | No. of cysts | 123.6 ± 8.3 | 91.0 ± 9.4 | 82.6 ± 4.6 | 103.8 ± 4.8 | 151.6 ± 3.4 | 4 |
| | FI (%) | 81.5 | 60.0 | 54.5 | 68.5 | 100 | |
| | Reaction | + | + | + | + | | |
| FN | No. of cysts | 0.4 ± 0.5 | 6.4 ± 1.9 | 12.4 ± 3.0 | 2.6 ± 1.3 | 157.2 ± 5.6 | 3 |
| | FI (%) | 0.3 | 4.1 | 7.9 | 1.7 | 100 | |
| | Reaction | - | - | - | - | | |

"+" denotes susceptible (FI ≥ 10%, compared to cv. Lee), and "-" denotes resistant (FI < 10%, compared to cv. Lee).

### 3.5. Koch's Postulates Test and the Host Range Experiment of the SBCN

Symptoms identical to those observed on naturally infested Chinese cabbage developed on inoculated Chinese cabbage. The plants were stunted, and the leaves turned chlorotic and yellow (Figure 4A). The roots grew slowly, and female nematodes were observed on the roots (Figure 4B). The second-stage juveniles, third-stage juveniles, fourth-stage juveniles and adults were observed in the roots of Chinese cabbage cv. Linglonghuang012. As observed in the field, the same nematode, SBCN, was reisolated from symptomatic roots of Chinese cabbage (Figure 4C–F), satisfying Koch's postulates.

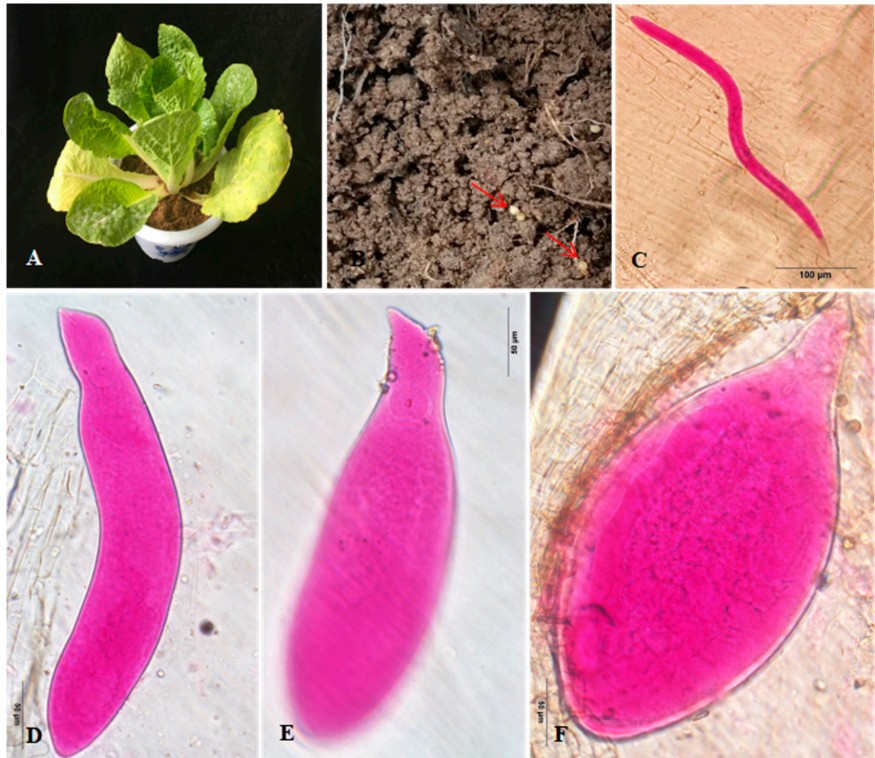

**Figure 4.** Symptoms of Chinese cabbage infested with SBCN. (**A**) Yellow leaves; (**B**) Cysts on roots; (**C**) Second-stage juvenile; (**D**) Third-stage juvenile; (**E**) Fourth-stage juvenile; (**F**) Female.

Fourteen crop cultivars from five plant families were evaluated as hosts for SBCN, which are major crops in the Bashang region. After 45 days of growth, all nematodes were extracted from individual plants and counted. Out of the 14 crop cultivars tested, SBCN reproduced on two cultivars but did not reproduce on the other 12 cultivars. All nematode stages were observed on Chinese cabbage (cv. Linglonghuang012) and cabbage (cv. Chunwang), $37.8 \pm 4.87$ and $13.8 \pm 2.39$ females were isolated, respectively, and all nematode stages were significantly different. $33.2 \pm 2.95$ J2s, $34.8 \pm 2.59$ third-stage juveniles and $6.8 \pm 1.48$ fourth-stage juveniles were found in roots of soybean (cv. Zhonghuang13), except for female nematodes. Only J2s were isolated from roots of canola (cv. Lvguan1), kidney (cv. Baifeng, Jiulibai and Baibulao), zucchini (cv. Jinghu12 and Zaoqing1), cucumber (cv. Zhongnong18), tomato (cv. Zhongza9), on average, $14.4 \pm 3.36$, $126.6 \pm 7.83$, $15.4 \pm 2.41$, $5.2 \pm 1.30$, $6.0 \pm 2.24$, $5.6 \pm 1.82$, $4.4 \pm 1.14$ and $11.4 \pm 2.30$ J2s were extracted, respectively. Potato (cv. Helan15), eggplant (cv. Junlang) and celery (cv. Yuhuang) were not infested (Table 4).

**Table 4.** Infection of *Heterodera schachtii* on different plants.

| Family | Tested Plants | Cultivar | No. of Second-Stage Juveniles | No. of Third-Stage Juveniles | No. of Fourth-Stage Juveniles | No. of Females |
|---|---|---|---|---|---|---|
| Brassicaceae | Chinese cabbage | Linglonghuang012 | 43.2 ± 4.15 b | 80.2 ± 5.50 a | 125.6 ± 12.34 a | 37.8 ± 4.87 a |
| | Cabbage | Chunwang | 21 ± 3.16 d | 19.8 ± 5.26 c | 9.8 ± 2.28 b | 13.8 ± 2.39 b |
| | canola | Lvguan1 | 14.4 ± 3.36 e | 0 d | 0 c | 0 c |
| Leguminosae | Soybean | Zhonghuang13 | 33.2 ± 2.95 c | 34.8 ± 2.59 b | 6.8 ± 1.48 b | 0 c |
| | | Baifeng | 126.6 ± 7.83 a | 0 d | 0 c | 0 c |
| | Kidney bean | Jiulibai | 15.4 ± 2.41 e | 0 d | 0 c | 0 c |
| | | Baibulao | 5.2 ± 1.30 f | 0 d | 0 c | 0 c |
| Cucurbitaceae | Zucchini | Jinghu12 | 6.0 ± 2.24 f | 0 d | 0 c | 0 c |
| | | Zaoqing1 | 5.6 ± 1.82 f | 0 d | 0 c | 0 c |
| | Cucumber | Zhongnong18 | 4.4 ± 1.14 f | 0 d | 0 c | 0 c |
| Solanaceae | Tomato | Zhongza9 | 11.4 ± 2.30 e | 0 d | 0 c | 0 c |
| | Potato | Helan15 | 0 g | 0 d | 0 c | 0 c |
| | Eggplant | Junlang | 0 g | 0 d | 0 c | 0 c |
| Umbelliferae | Celery | Yuhuang | 0 g | 0 d | 0 c | 0 c |

Each seedling was inoculated with 800 *Heterodera schachtii*-second stage juveniles (J2s) and five plants were sampled per cultivar after 3, 8, 15, 30 and 45 days of incubation. Data are the mean ± SD. Different lowercase letters in the same column indicate significant differences among different hosts ($p < 0.05$).

## 4. Discussion

In this study, four populations (ZM, KM, CB and FN) of SCN and one population (ZZ) of SBCN were isolated and identified from the soil in the Bashang region of Hebei Province by the floating separation method. The densities reached 57, 41, 103, 31 and 94 cysts per 200 cc of soil on average, respectively. Steady expansion of the distribution of SCN throughout the United States and Canada, and SCN almost certainly will continue to spread [35,36]. SBCN is listed as a quarantine nematode by China. Unfortunately, it was found on sugar beet in Xinyuan County, Xinjiang [20]. During our field investigation, cyst nematodes were found on Chinese cabbage, with stunted plants and yellow leaves. This is the typical symptom of cyst nematode infestation. A new disease of Chinese cabbage caused by SBCN was determined according to Koch's postulates, which is the first report of SBCN on Chinese cabbage in Zhangbei County, Hebei Province, China. All nematode stages of SBCN were observed inside or on roots of Chinese cabbage (cv. Linglonghuang012). SBCN has been previously reported on Chinese cabbage in South Korea [21]. They will be a major threat in the Bashang region. However, information on the occurrence and distribution of SCN and SBCN was limited in the Bashang region of Hebei Province. Therefore, there was an urgent need to determine the occurrence and distribution of SCNs and SBCNs. Their presence as per our findings will be a major threat in the Bashang region and a call for strict measures in the control and manage SCNs and SBCNs successfully.

The morphological characteristics of the populations (ZM, KM, CB, FN and ZZ) were observed and measured. The ZM, KM, CB and FN populations observed in the study were consistent with those mentioned in the references about other populations of SCN [3,37]. The ZZ population was almost consistent with the populations from Xinjiang [20], Turkey [38], South Korea [39] and the USA [17]. Furthermore, molecular analyses of rDNA-ITS confirmed that the ZZ population clustered in a branch with the Xinjiang and Belgian populations of SBCN, with high similarity of 99.79–100%. The specificity analysis showed that six positive bands of 477 bp fragments and four positive bands of 922 bp fragments were observed by SCN-specific primers and SBCN-specific primers, respectively [28,31]. It is a rapid, reliable and simple protocol using a species-specific SCAR-PCR assay to detect SCN and SBCN.

To reduce experimental errors, the SCN identification procedure was optimized in this study. The diseased field soil was used to identify races of SCN [40]. However, there

were always mixed races, so the results could not fully represent the actual situation. When resistant varieties are planted for many years, races of SCN vary [41]. The virulence of SCN populations is shifting and overcoming resistance [42], so the variants can be classified, and plant breeders are able to identify resistance to specific races. It is necessary to use different resistance sources and identify novel resistance sources for SCN management. In Hebei Province, SCN races 1, 2 and 5 have been found in Handan, Gaocheng and Cangxian, respectively [7]. In this experiment, SCN race 4 was found in Zhangbei, Kangbao and Chabei of Zhangjiakou, and SCN race 3 was found in Fengning of Chengde. However, ten cysts and six juveniles of SCN were isolated in one soil sample from Zhangjiakou, whose race was not clear [6]. To the best of our knowledge, this is the first report about SCN in the Bashang region of Hebei Province.

Chinese cabbage (cv. Linglonghuang012) and cabbage (cv. Chunwang) were suitable hosts for SBCN, $37.8 \pm 4.87$ and $13.8 \pm 2.39$ females were isolated, respectively, and all nematode stages were significantly different between them. Plants of Brassicaceae are suitable for infestation and reproduction of SBCN [22,43,44]. Canola (cv. HyClone 940, Caliber 30, and Westar) were grown in the greenhouse, which was inoculated with 500 eggs. After 53 days of incubation, there was an average of 39, 20, and 30 females for each respective cultivar [17]. A total of 194 J2s and 26.14 cysts were found on canola (cv. Huiyou808) [34]. Second-stage juveniles, females and cysts were found on oil seed rape (cv. Deyou 6), after inoculation with 500 J2s [20]. In our study, only $14.4 \pm 3.36$ J2s were observed on canola (cv. Lvguan1), no females were found, the differences may be due to the different varieties, and further experiments should be conducted to confirm these results. On average, $33.2 \pm 2.95$ J2s, $34.8 \pm 2.59$ third-stage juveniles and $6.8 \pm 1.48$ fourth-stage juveniles were found in roots of soybean (cv. Zhonghuang13), but no females were observed, while $113.4 \pm 9.56$ J2s and 0 cysts were observed in the host range experiment [34]. Only second-stage juveniles were isolated from roots of canola (cv. Lvguan1), kidney (cv. Baifeng, Jiulibai and Baibulao), zucchini (cv. Jinghu12 and Zaoqing1), cucumber (cv. Zhongnong18), tomato (cv. Zhongza9), which indicated that SBCN could only infest but could not complete its life cycle. SBCN can be controlled biologically in highly infested soils of sugar beet rotation using resistant varieties of oil radish as a green crop. Resistant plants stimulate infective juveniles to invade roots but prevent them from penetrating to complete the life cycle [44]. Potato (cv. Helan 15), eggplant (cv. Junlang) and celery (cv. Yuhuang) were not infested in this study. The three cultivars tested were nonhosts, and the nonhost crops can be planted in infested fields without the concern of increasing SBCN populations. Resistant cruciferous plants used as intercrops for trapcropping inhibit further nematode development, and growing trap crops on severely infested soils can decrease the pathogen population density [43]. Even though different management strategies are available to manage SBCN infestations, the most effective methods in infested fields are three- and/or four-year nonhost crop rotation and the use of *H. schachtii*-resistant cultivars for the nematode management. This study also provided new information concerning the potential use of nonhosts as a control method.

## 5. Conclusions

Four populations of SCN and one population of SBCN were isolated and identified in the Bashang region of Hebei Province; of them, the ZM, KM and CB populations were all SCN race 4, while the FN population was SCN race 3. A new disease on Chinese cabbage caused by the ZZ population of SBCN was determined according to Koch's postulates. Chinese cabbage (cv. Linglonghuang012) and cabbage (cv. Chunwang) were suitable hosts, while celery (cv. Yuhuang), potato (cv. Helan 15) and eggplant (cv. Junlang) were nonhosts. To the best of our knowledge, this is the first report of SCN and SBCN in the Bashang region of Hebei Province.

**Supplementary Materials:** The following supporting information can be downloaded at https://www.mdpi.com/article/10.3390/agronomy12092227/s1, Table S1: Soil samples collected from the Bashang region of Hebei Province; Table S2: Morphometrical comparisons of cysts and second-stage juveniles (J2s) of SCN between Bashang populations and other populations; Table S3: Morphometrical comparisons of cysts and second-stage juveniles (J2s) of SBCN between ZZ population and other populations.

**Author Contributions:** Conceptualization, D.L., D.P. and H.P.; Methodology, Y.W., H.S. and H.P.; Investigation, Y.W., H.P., Y.Z. and Y.L. (Yunqing Li); writing and original draft preparation, Y.W.; Editing, S.L., Y.L. (Yaning Li) and D.P. Supervision, D.P. and H.P. All authors have read and agreed to the published version of the manuscript.

**Funding:** This research was supported by the National Key R&D Program of China (2021YFD1400100), the National Natural Science Foundation of China (31972247) and the Science and Technology Innovation Project of Chinese Academy of Agricultural Sciences (ASTIP-02-IPP-15).

**Institutional Review Board Statement:** Not applicable.

**Informed Consent Statement:** Not applicable.

**Data Availability Statement:** The original contributions presented in the study are included in the article. Further inquiries can be directed to the corresponding author/s.

**Conflicts of Interest:** The authors declare no conflict of interest.

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
