# Peer review of "Investigation and Identification of Cyst Nematodes in the Bashang Region of Hebei, China"

_agronomy, doi:10.3390/agronomy12092227_

Round 1

Reviewer 1 Report

The manuscript ID: agronomy-1896590 deals with investigation and identification of cyst nematodes in the Bashang region of Hebei, China. It addresses the occurrence, distribution, races of Heterodera glycines (SCN) and hosts of Heterodera schachtii (SBCN) in the Bashang region to offer a reference for their management. It is quite good for publication, as it covers the different topics from taxonomy and species/races identification, pathogenicity, and tools for their management, besides showing their host range. As the most effective methods in CN-infested fields are nonhost crop rotation and the use of resistant cultivars, I’d propose that the authors state three- and/or four-year crop rotations for the nematode management. Titles of tables should be more explicit and informative, in Table 3, Heterodera glycines should replace SCN; in Table 4 you should add as a footnote that each seedling was inoculated with 800 Heterodera schachtii-second stage juveniles (J2s). I’d also suggest that the English of the article be proofread, as there are small errors or misprints that can be improved (e.g. BSCN instead of SBCN in line 109).

Therefore, I would suggest accepting the MS after minor revision.

Author Response

We are grateful thank you for time spend reviewing this article and for your recognition and support.

Author Response

We thank the expert for helpful comments that will greatly improve our manuscript. We have tried to do our best to respond to the points raised. The revisions are listed below. 
